# BIM and Digital Twin for Developing Convergence Technologies as Future of Digital Construction

Samad M. E. Sepasgozar [1], Ayaz Ahmad Khan [2], Kai Smith [1], Juan Garzon Romero [1], Xiaohan Shen [1], Sara Shirowzhan [1,*], Heng Li [3] and Faham Tahmasebinia [4]

1   School of Built Environment, The University of New South Wales, Sydney, NSW 2052, Australia
2   Australian Research Centre for Interactive and Virtual Environments (IVE), UniSA Creative, University of South Australia, Adelaide, SA 5000, Australia
3   Department of Building and Real Estate, The Hong Kong Polytechnic University, Hong Kong 999077, China
4   School of Civil Engineering, The University of Sydney, Darlington, NSW 2006, Australia
*   Correspondence: s.shirowzhan@unsw.edu.au

**Abstract:** The construction industry is slow to adopt new technologies. The implementation of digital technologies and remote operations using robots were considered farfetched affairs and unbelievable approaches. However, the effect of COVID-19 on clients and construction companies put high pressure on construction managers to seek digital solutions and justified the need for remote operating or distant controlling technologies. This paper aims to investigate the state of play in construction technology implementation and presents a roadmap for developing and implementing required technologies for the construction industry. The COVID-19 disruption required new methods of working safely and remotely and coincided with the advent of advanced automation and autonomous technologies. This paper aims to identify gaps and 11 disruptive technologies that may lead to upheaval and transformation of the construction sector, perhaps in this decade. A road map for technology implementation can be helpful in developing business strategies at the organizational level as a theoretical measure, and it can facilitate the technology implementation process at the industry level as a practical measure. The roadmap can be used as a framework for policymakers to set industry or company strategies for the next 10 years (2030).

**Keywords:** digital twin; BIM; construction industry; virtual reality; Industry 4.0; COVID-19; digital solutions; convergence technology

## 1. Introduction

Major industries are currently going through a digital transformation known as the Fourth Industrial Revolution (IR-4.0), driven by exponentially increasing computing power and abundantly available electronic data [1]. For some time, rising digital technology adoption has been reported in numerous market reports and industry surveys [2]. Recently, however, The Mckinsey Global Institute noted a dramatic increase in digitization due to the COVID-19 pandemic, which has led to a doubling of revenue compared to pre-pandemic estimates [3]. Reliance on mobile and cloud computing has skyrocketed in the wake of COVID-19 and will continue to do so for several years as industrialized countries stage an economic recovery [4]. To that end, cyber-physical systems' time/cost-saving potential offers a promising pathway to a sustainable economic recovery [5].

Although COVID-19 has led to many companies exiting the marketplace [6], including the closure of industry disrupter Katerra [7], IR-4.0 initiatives coupled with enhanced remote work facilitation, better supply chain connections, and transparency in operational works will probably be the driving force behind the companies which will emerge stronger in the aftermath of the pandemic [8]. New business models centered on digitalizing products, processes, and materials will redefine the construction industry. A recent survey of construction industry vendors, to cite an example, forecasts digital Twin (DT) as a mature

evolution of Building Information Modeling (BIM) to enable the transformation of the sector [8]. As a result of this digital technology adoption, the architecture, engineering, and construction (AEC) market size is expected to surge from $7188 million in 2020 to $15,842 million by 2028 at a compound annual growth rate (CAGR) of 10.7% from 2021 to 2028 [9].

This mega-trend of advancing digital technologies and innovative solutions during the pandemic gives some hope to the construction industry that surviving COVID-19 may lead to an increase in productivity in the future [10]. However, it brings some uncertainty to smaller and medium-sized companies as to whether they can afford expensive digital technologies or develop innovative in-house solutions to continue their business [6].

Similar to other businesses, digital technologies are essential for health testing, contact tracing, and monitoring workers' safety on construction sites. At the same time, cloud and mobile computing systems enable remote coordination of designers and construction workers, and they can quickly optimize and restructure construction supply chains [11]. Despite the widespread availability of all these technologies, enabling telework and remote operation on construction sites, the construction industry has, until COVID-19, had much less experience with digitization than consumer goods manufacturing and other sectors [2]. Telework technologies or robots may eventually change the nature and procedure of construction operations and management permanently and radically [12]. However, there are currently many ambiguities and a lack of investigation into future directions and what construction workers should expect in the latter half of the 21st century [12]. The answer to this question is fundamentally important since it will be the basis of strategy development and investments for upskilling the workers and preparing economic infrastructure for the future. This paper intends to take the first step towards mapping the future of digital technologies based on needs identified during COVID-19, contributing towards filling the circularity gap and climate-resilient innovations and technologies in the construction context. To unleash this black box and holy grail in the construction domain, the future of construction innovation and technology (FOCIT) literature is studied in this research. A few key concepts should be reviewed before conducting a FOCIT search within the literature. Firstly, disruptive technologies refer to technologies that have the potential for major upskilling or revolutionary changes in construction processes, procedures, and operations at the time [13]. Next, emerging technologies refer to technologies that are mature at the time but have not been widely used by then, or the effect of the technology on construction processes and needs have not been clarified [13]. Finally, convergent technologies in construction refer to the amalgamation or integration of technologies in a novel way that may affect the industry or operation processes and can be counted as disruptive [13].

The two important digital technologies for sharing and managing information that paves the way for "converging technologies" in construction are BIM and DT [14]. While BIM is considered a "single source of truth" for geometrical and non-geometrical information in the construction industry, DT is perceived as a "single version of the truth" encompassing BIM's physical and dynamic representation, particularly with the integration of blockchain [15]. BIM is a digital representation of the physical and functional characteristics of a building, which is created using specialized software such as Revit, and Archi CAD, among others [12]. BIM models include information about the building's design, construction, and operation and can be used for a variety of purposes, such as design coordination, construction planning, and facility management [16]. The BIM models can be used throughout the entire building lifecycle, from the design phase to the construction phase, to the operation and maintenance phase of a building [17]. A DT is a virtual replica of a physical object or system, such as a building or facility [18]. A DT is typically created using data collected from sensors and other sources, such as BIM models. It is used to represent the current state of the physical object or system. The DT is a dynamic, digital representation of the physical object that can be used to simulate, analyze, and optimize the performance, operations, and maintenance of the physical object or system [10]. These two technologies are paving the way for other digital technology to converge and provide necessary and concomitant solutions for the construction industry.

The concomitant and burning research objectives are (i) to identify the FOCIT literature focusing on innovation, in particular, emerging, disruptive, or convergent technologies; (ii) to consider and expand upon the likely strategic horizons for the construction industry in light of foreseeable risks; challenges and threats in the latter half of the 21st century; (iii) to identify disruptive technologies in the construction sector that will emerge in the coming decade; (iv) to analyze the current situation of integrated technology use, lifecycle application, and sustainability concern for information management in the construction industry; (v) to discuss what is the future of visual technologies and what it takes to achieve it; and (vi) to establish a set of directions for future investigations on technology implementation in the construction industry.

## 2. Research Methodology

Following paradigms of epistemology, this study adopted a multistep methodological approach, including the systematic process for each step, as shown in Figure 1 [19]. This approach structures the output knowledge in sequential steps and further increases the findings' credibility and reliability [20]. The steps include data collection, selection, and analysis with procedures and principles of performing a task.

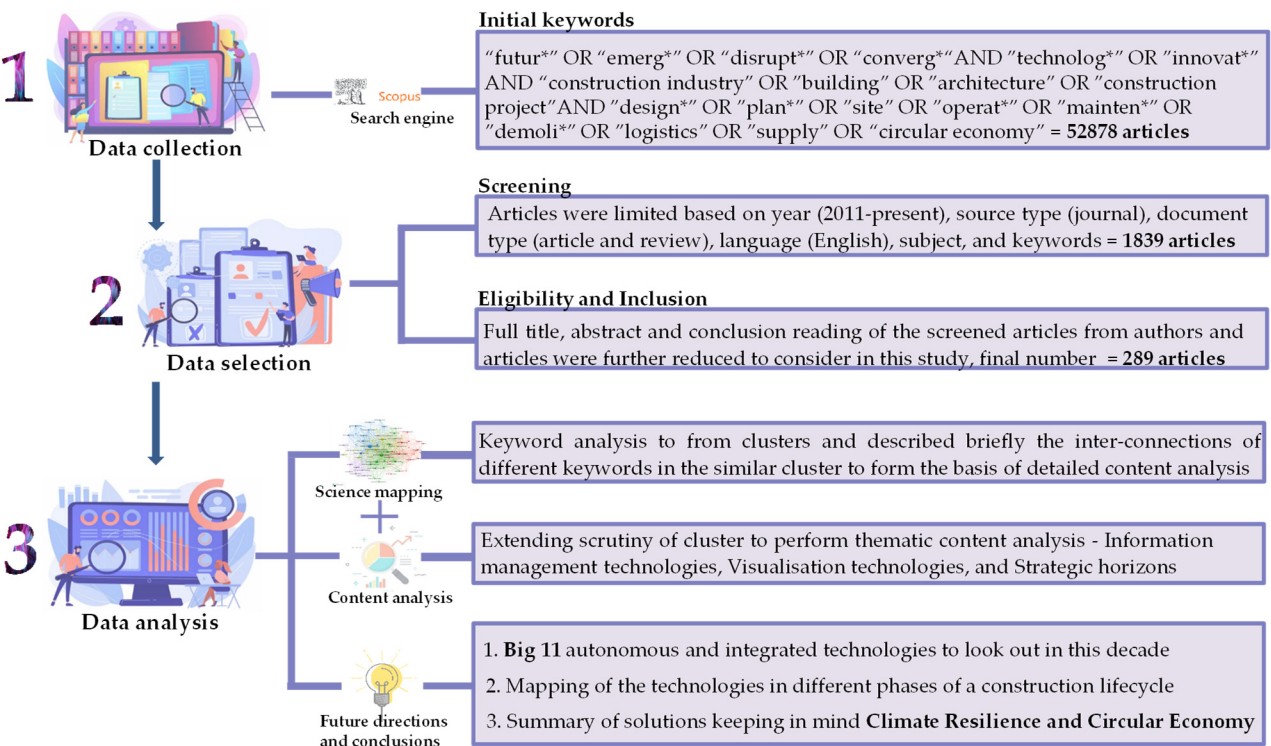

**Figure 1.** The multi-stage methodological framework for the study.

Steps 1 and 2 follow the article retrieval process using preferred reporting items for systematic reviews and meta-analysis (PRISMA) protocol [14]. Initially, a brainstorming session is conducted among authors to conclude the keywords for the article's retrieval. Citing the vastness of the AEC industry and the nature of the study, the keywords were extracted and put into the database. The database used for the article retrieval process was Scopus, an authoritative search engine covering a wide pool of publications. Also, as compared to the Web of Science, Scopus covers more relevant journals and publications [19].

Keywords were input in the Scopus database using the TITLE-ABS-KEY as follows: "future*" OR "emerg*" OR "disrupt*" OR "converg*" AND TITLE-ABS- KEY "construction industry" OR "building" OR "architecture" OR "construction project" AND TITLE-ABS-KEY "design*" OR "plan*" OR "site" OR "operat*" OR "mainten*" OR "demoli*" OR "logistics" OR "supply" OR "circular economy". A total number of 52,878 articles were

extracted at this step. In step 2, the screening is done based on the following parameters: (1) year limitation of the articles from 2011 to 2022 (October) as the focus of the study was limited to articles from the last decade, (2) document type and source type to article or review and journal, (3) published in English, and (4) relevant subjects and keywords. The number of articles that remained after this stage was 1839. Furthermore, the title, abstract, and conclusion were read by different authors, and 1531 articles were removed in the eligibility phase as their focus or application was not directly related to the scope of this study. Finally, 289 articles were included for scientometric keyword analysis and critical content review for the study, as shown in Figure 1.

Step 3 of the data analysis starts with science mapping the 289 articles retrieved in the previous stage. In this case, the VosViewer tool is used for science mapping to highlight the important keywords of FOCIT literature. Figure 2 shows the keyword science mapping of the 289 articles reflecting different clusters. Next, scrutiny of the clusters is performed for critical content analysis based on the themes of information management technologies, visualization technologies, and strategic horizons. Further, a discussion on Big 11 autonomous and integrated technologies is laid out that will be the key for the construction industry in this decade. Further, mapping these technologies is carried out for different stages of a construction project integrating the circular economy (CE) principles of resource loops. Finally, key solutions, future trends, and strategies for technologies in the FOCIT domain contemplating Climate Resilience are reflected.

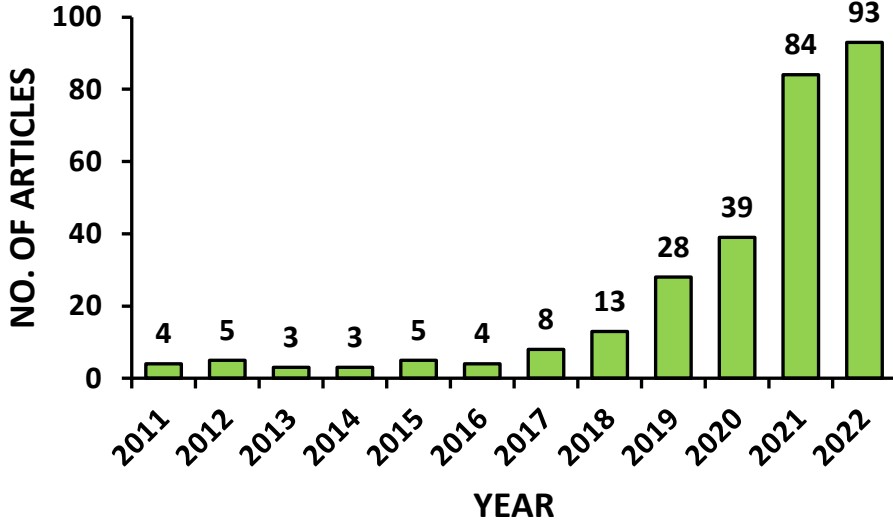

**Figure 2.** Year-wise (January 2011 to October 2022 distribution of retrieved articles.

### 3. Bibliography Results

#### 3.1. Annual Publication Trend of the FOCIT Literature

A total of 289 articles were published based on the search criteria in the last decade. The small yet significant number probably highlights the relevance and shift of the AEC industry towards modern techniques, whether in design and planning, construction, or operations stages. This reinforces the timely need for this study to seek the individual effect of the technology or integrative approaches, generating a holistic overview considering the FOCIT literature. The span of the studies is between January 2011 to October 2022, as shown in Figure 2. An exponential rise in the studies on the subject topic is evident in the last three years, with skyrocketing numbers in 2021 to 84 and 2022 to 93 and counting. The substantial growth in the articles is a result of the emergence of modern techniques utilization in all stages of the construction industry [2]. The effect of IR-4.0 in the last five years, which is deemed by the researchers as the onset of modern and digital technologies, is reflected in the rise of publications in those years. This trend is solidified by the assumptions of Sepasgozar [2] that the recipe for success in the construction industry and the upheaval of low-productivity images will be achieved by integrating IR-4.0 techniques.

### 3.2. Contribution of Journals in FOCIT Literature

The importance of highlighting the journal distribution reflects the quality of studies in the FOCIT literature. The studies were published in high-impact journals, as shown in Table 1, which shows the significant effect of the research scope of the subject matter. Due to the overwhelming nature of journal distribution, only the top 10 journals with the maximum number of articles are shown in Table 1. The most influential journals are AiC, ECAM, Buildings, CI, and JCLP, and they are some of the highest-ranked journals in the AEC and construction and engineering management (CEM) research areas [20]. The superiority of AiC and CI is justified as the adoption of digital technologies automates and innovates the construction paradigm [12]. Further, many studies utilized these digital technologies to propose cleaner and energy-efficient methods, which marks the importance of journals like JCLP. Other significant areas such as construction management, use of IT in construction, sustainable aspects, and construction engineering methods solidify journals such as ECAM, ITCon, Sustainability, and JME, respectively. In a nutshell, this table of journals provides researchers the relevant information for their scholarly submissions.

**Table 1.** Journal contribution based on articles published (top ten).

| Name of Journal | Number of Articles |
| --- | --- |
| Automation in Construction (AiC) | 41 |
| Engineering, Construction, and Architectural Management (ECAM) | 32 |
| Buildings | 25 |
| Construction Innovation (CI) | 21 |
| Journal of Cleaner Production (JCLP) | 19 |
| Journal of Information Technology in Construction (ITCon) | 13 |
| Sustainability (Switzerland) | 12 |
| Journal of Construction Engineering and Management (JCEM) | 12 |
| Journal of Building Engineering (JBE) | 11 |
| Journal of Management in Engineering (JME) | 10 |

### 3.3. Research Instruments Utilized in the FOCIT Literature

To deliver a guide for future studies and the quality of research, the utilized data instruments or research methods are listed in Table 2. The methods used are common in construction domain research and entail surveys/questionnaires, interviews, experiments, workshops, case studies, mixed methods, and review studies [21]. The table signifies the different approaches utilized by the researchers for data collection and presentation of their studies. The significant number of review articles reflect the urge of researchers to provide a holistic landscape of individual technologies or a combination of few in the construction industry. For instance, Sepasgozar [14], in his review study, unleashed the variation between the "digital twin" and "digital shadow", which is a considerable effort citing the ambiguity in these two terms.

**Table 2.** Distribution of research instruments in the FOCIT literature.

| Research Instruments | Number of Articles |
| --- | --- |
| Survey/Questionnaire | 32 |
| Interviews | 20 |
| Simulation/Modelling | 48 |
| Workshops | 21 |
| Case Studies | 32 |
| Mixed Methods | 67 |
| Reviews | 69 |

Further, the development of prototypes, scenarios, and samples is common in the construction industry. However, this effectiveness is measured through well-performed

simulation and modeling methods. Out of 289 studies, at least 48 studies in the FOCIT literature applied simulation or modeling approaches to present their results.

### 3.4. Science Mapping of the Relevant Keywords from FOCIT Literature

The keyword cluster and their connections show the articles' underlying interests, reflecting promising integrative opportunities with other keywords in the same or other clusters. Figure 3 below maps the keywords of the 289 studies from the FOCIT literature using the Vos-Viewer science mapping tool [22]. It is clear from the figure that technologies are utilized in various domains of construction. Different clusters and their characteristics are described as follows:

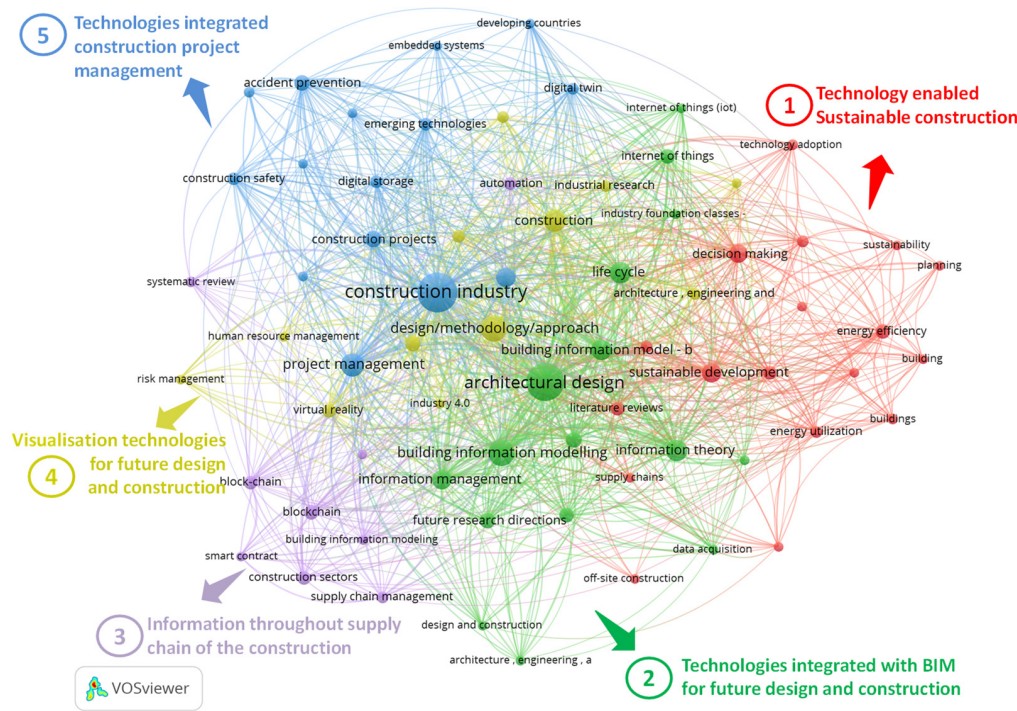

**Figure 3.** Science mapping keyword analysis of retrieved articles.

Cluster one in "Red" is "Technology enabled Sustainable Construction." The quest for sustainable construction is significant in the AEC industry. Words like energy-efficient construction and green construction methods are often used simultaneously or in place of sustainable construction. The cluster focuses on different technology adoption to optimize sustainability in construction. The studies utilized artificial intelligence (AI) techniques to develop a proof of concept, theoretical frameworks, and case studies to enhance sustainability in construction projects. For instance, in a recent study, Debrah et al. [23] reviewed AI in green buildings. They categorized the research domain into fuzzy rule and knowledge discovery, intelligent optimization, building automation systems, and big data and data mining.

Further, the rise in sustainable construction methods, such as offsite and modular integrated construction, was seen in developing and developed countries to culminate in affordable housing issues [24]. Wang et al. [25] synthesized the adoption of digital technologies in the offsite construction methods for better-optimized results. The future will include circular economy principles with sustainable construction to develop more energy-efficient buildings and net zero emissivity [26].

Cluster two in "Green" is "Technologies integrated with BIM for future design and construction". Over the last two decades, BIM has revolutionized the design and construction of projects in the AEC industry; however, integrating other technologies makes the design process more automated and iterative. Computational design methods and

their subcategories, namely parametric design, algorithmic design, and generative design, give designers and architects the leverage to have multiple solutions for a particular problem [27]. The tools supporting these design methods operate on an AI technique that not only aids in offering manifold design options but also stipulates the best option in terms of energy and space usage. Further, the utilization of digital technologies and methods such as the Internet of things (IoT), big data, and AI has made it possible for a BIM model to become a DT, that is, a digital replica of a construction project delivering the benefits throughout the life cycle of a project [14]. Further, designing buildings in virtual worlds such as the Metaverse will engage clients and stakeholders more collaboratively, thus reducing the shield of information sharing [28]. Finally, as the world is moving towards applying CE principles in their businesses, the regeneration and reuse of virgin materials will accelerate the "bin to BIM" process [29].

Cluster three in "Purple" is "Information throughout the construction supply chain". The information management between various stakeholders in construction projects has always been a concerning issue. The construction industry is plagued with issues of information sharing that result in low productivity, inefficient compliance, poor regulations, inadequate collaboration, and disorganized payment practices. Recent advances in distributed ledger technologies (DLT), also called blockchain technology (BCT) prove to be a solution to these challenges [30]. BCT's expected benefits are increased collaboration, disintermediation, quick immutable process, low human error, traceability and provenance, and workflow improvements, among others [31]. Various studies highlighted the effectiveness of BCT in the construction industry, especially in generating BIM-integrated smart contracts, information sharing in onsite assembly tasks of modular construction, BIM-BCT-based design collaboration issues, reducing construction disputes, and integrated project delivery [32]. Few other studies have integrated different digital technologies to propose a proof of concept to support information sharing throughout the supply chain of a construction project. Li et al. [33] proposed a service-oriented BCT-enabled IoT-BIM platform (BIBP) in modular construction supply chain management. The proposed architecture can deliver latency in storage in a privately secured IoT network. In another study, Lee et al. [34] integrated DT and BCT to provide traceable data transactions in near real-time for a prefabricated project. On the market side, many decentralized autonomous organizations (DAO) are emerging to control the tasks related to BCT along with numerous decentralized applications (DApps) [15]. However, a few challenges, such as data authentication, adoption readiness, change resistance, skill shortage, malicious attacks, and connectivity, still prevail that need to be prevented from availing long-term paybacks from BCT in the construction industry [30].

Cluster four in "Yellow" is "Visualization technologies for future design and construction". Recent advances in visualization technologies in the construction industry have systematized the design and planning construction management process and improved the wicked image of the construction industry in terms of the health and safety of the construction workers. From BIM 4D planning to integrating extended realities (XR) such as virtual reality (VR), augmented reality (AR), and mixed reality (MR), the tasks at different stages of a construction project have improved holistically [35].

Cluster five in "Blue" is "Technologies integrated construction project management". According to the project management institute (PMI), project management is the utilization of knowledge, skills, tools, and techniques to deliver projects that are valuable to people [36]. This cluster focuses on different methods and technologies used in construction projects to make them significantly productive and profound. A successful project focuses on time, cost, quality, and productivity measures and ensures health and safety throughout the delivery of a project.

## 4. Content Analysis and Critical Review

### 4.1. Strategic Horizons

In developed and developing countries alike, the construction industry forms the backbone of the national economy. It is pivotal in providing the infrastructure necessary to improve the standard of living and well-being of citizens. Increasingly, well-being is subject to various existential threats associated with climate change, environmental destruction, and rapid urbanization [37]. Moreover, providing equitable access to affordable housing, healthcare, education, and public services has become a pressing concern of governments worldwide [37]. Dwindling natural resources and a globalized marketplace also means a balance must be struck between the cost of providing these social goods and their productive efficiency in the long term. Accordingly, the meaning of efficiency has, in the second decade of the 21st century, expanded considerably to include a complex variety of novel attributes such as environmental sustainability, social equity, financial rigor, design merit, and well-being [38]. With that in mind, how governments plan, procure, operate, and maintain public works is shifting in lockstep with community expectations to prefer large-scale government projects which are fully digitized and fit for purpose in an envisaged CE practice [26]. While consensus on what constitutes a holistic CE practice remains elusive [38], it is reasonable to say its achievement will probably involve substantial industrial reform aided by digital technology and automation and the reskilling of the existing labor force.

In any event, the construction industry will likely remain characterized by its heavy reliance on manual labor and small to medium enterprises well into the latter half of the 21st century. As per Myers D, the decentralized, diffuse nature of the sector will most probably guarantee its ongoing economic importance [38]. Contrary to popular science fiction, mass automation with virtually no worker involvement is inconsistent with the long-term interests of modern governments [39] because construction remains among the largest employers of working-class male youths and, depending on the definition used, includes a wide range of professional services, high-tech manufacturers, and primary industries in addition to building activity. In most jurisdictions, the industry tends to be heavily monitored and regulated by national governments for several reasons [38]. Firstly, construction projects usually constitute the single largest purchase an individual or incorporated entity is likely to invest in and must be safeguarded by the rule of law. Secondly, governments are usually the largest consumers of construction industry services. The imperative to deliver public works according to certain axiomatic principles such as value for money, occupational health and safety, and minimizing environmental damage requires stringent oversight of the sector and its practices. And thirdly, the sector forms the centerpiece of economic policy and is a means for state intervention in the marketplace when economic efficiency falters or non-market events, such as natural disasters, armed conflict, and pandemics, disrupt economic growth. This is unlikely to change in the latter half of the 21st century. When such events occur, governments will enact fiscal measures, such as additional spending on new transportation infrastructure, public housing, new healthcare facilities, and militarization, if necessary, to stimulate industrial activity and return the economy to a productive equilibrium. Governments, as one of the largest consumers of construction industry services in any given economy, will also seek to enhance productivity in terms of the economy's ability to optimally use scarce resources [40] by adjusting public policy settings, creating tax incentives to adopt technology, directing resources to the education system, and funding research.

The construction industry itself is notoriously conservative in that it tends to regard new technologies with circumspection and is not necessarily given to passive conformism when faced with unpalatable adjustments to public policy settings announced by the government [41]. Nevertheless, a reckoning on tackling intensifying economic and social headwinds is fast approaching, and radical government intervention in response to climate change and the like is inevitable. Decades of debate, subsidies, and all manner of financial incentives [42] have failed to produce the kind of efficient allocation of resources needed

to avert disaster because the construction industry has, for centuries, evolved as a project-based economy that is, ironically, extremely sensitive to economic shocks, while uniquely inelastic to sudden changes in demand or disruptions to the supply chain. Put simply, houses and hospitals are, generally, not like cars and denim jeans, items beholden to the caprice of fashion and whim.

However, the breathtakingly fast construction of the Huoshenshan Hospital in Wuhan [43] in response to the outbreak of COVID-19 in that city is a compelling and instructive example of what construction could look like in an era set to be defined by rolling global crises. In those circumstances, industrialized countries could become leading exporters of critical infrastructure worldwide. This would be a game-changer regarding their influence and ability to project soft power on the world stage. From that point of view, it seems China may have already established a competitive advantage due to its centrally planned command economy and decades of investment in building its high-tech manufacturing capacity and labor force. Consider, for example, the China communist party's (CCP's) ambitious response to Industry 4.0, Made in China 2025 [44]. Meanwhile, the United States (US) administration struggled to seek approval for its "Build Back Better Bill", which, notwithstanding a herculean sum dedicated to climate change, contained no penalties for failing to take meaningful action whatsoever [45]. Other countries which have focused instead on raw material exports, such as Australia, Brazil, and Canada, may struggle to compete and will have to commit heavily to national innovation programs to expand their manufacturing sectors, which could take decades to catch up. In that case, the spectrum of advanced technologies covered in this article may need to be marshaled creatively to harness the full potential of construction manufacturing.

Commentators have, for some time, predicted a shift to high-tech construction manufacturing to minimize waste, deliver projects safely and efficiently, and realign industry with the changing makeup of the labor force. The concept of off-site prefabrication has been around for several hundred years [46], and it was an essential element of the British colonial project, for example. Moreover, interest in this method has waxed and waned since then, particularly in the US, Australia, the United Kingdom, Singapore, and Malaysia [46]. In its modern iteration, construction manufacturing has become a proxy for prefabrication under IR-4.0, the leading paradigm for industrial transformation [12].

IR-4.0 refers to a German national innovation initiative for a planned fourth industrial revolution aiming to secure Germany's position as the leading manufacturer of high-tech consumer goods [47]. The initiative is composed of a complex network of government agencies, scientific institutions, and research projects established in central Germany after the 2011 Hanover Fair [1]. Internationally, IR-4.0 has become equated with the adoption of certain ground-breaking technologies such as big data, AI, cloud computing, robotics, the IoT, and BIM. Essentially, the object of IR-4.0 (setting aside its socio-political attributes for the moment) is achieving a virtuous cycle of horizontal and vertical integration in the controlled manufacturing environment of a Smart Factory. Figure 4 below illustrates this argument diagrammatically.

This could, in principle, overcome several operational risks in modern construction, not least among which is the gap between suppliers of raw materials and the end users of their products. Given the sheer scale and complexity of construction projects, a sophisticated platform for managing and visualizing vast amounts of data as well as the human–machine interface, would be needed to achieve the degree of interoperability, precision, and standardization demanded. Figure 4 illustrates a possible model based on Osterrieider's et al. [48] four-layered smart factory concept.

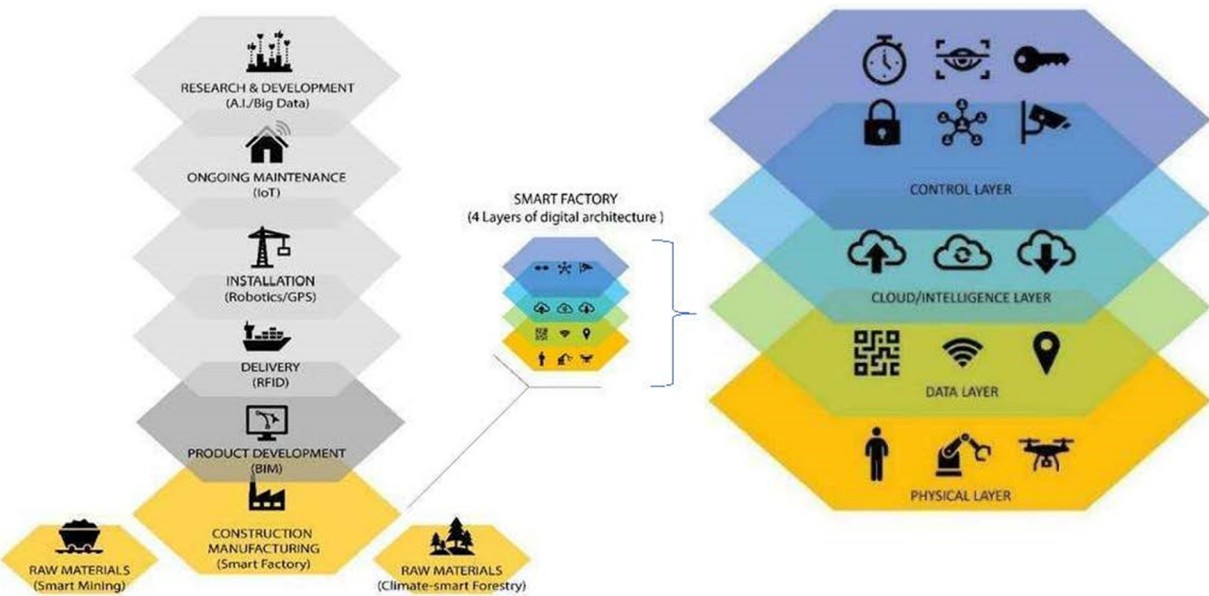

**Figure 4.** Horizontal and vertical integration in consumer goods manufacturing.

In Figure 4, the four-layered digital ecosystem of a Smart Factory concept comprises integrated information technologies applicable to city shaping, construction, and consumer goods manufacturing in equal measure. It cannot be stressed enough that academics and practitioners must bear in mind that IR-4.0 is an initiative developed progressively in Germany, where high-tech consumer goods manufacturing is well established and a key component of gross domestic product. Furthermore, even in Germany, the concept has achieved only modest penetration in the construction sector thus far.

Noting this vacuum, Oesterreich and Teuteberg [48] identified several directions for future research, not least among them was the societal impact of digital transformation under the scheme. Much of the international literature on IR-4.0 tends to focus on the high-tech devices needed to achieve the so-called Fourth Industrial Revolution objectives instead of its socio-political objectives or the regulatory means to achieve them, which is understandable given the concept's high-tech appeal [49]. However, society's back door must not be left open to the possibility of corporate predation and other unintended consequences in the sector by neglecting important considerations such as strict regulation of the labor market and the natural environment, especially when the going gets tough and when climate change starts to bite.

It is no secret that construction workers are already among the most burdened by serious accidents on the work site, so digital transformation must be carried out to enhance worker safety along with a steadfast belief in the dignity of work. As construction processes evolve, it is likely that a range of new risks to workers' safety will arise alongside novel methods of coordination and assembly should building in a controlled manufacturing environment become mainstream; consequently, valuable taxpayer funding will need to be dedicated to preparing a skilled workforce, underpinned by robust regulatory regimes and industry standards moving forward. Ideally, national governments should become proactive in setting the agenda for technological advancement in this area. In this regard, Xu's et al. [50] article calling attention to the Singapore government's robust interventions in the marketplace is instructive.

Finally, there's the impact of increased automation and mechanization on indigenous and remote communities living near vital mineral deposits needed to produce the advanced technology for transitioning to a low-carbon economy, particularly in the global south; their long-term interests must be prioritized in exchange for access to their land. On that basis, academics such as Oti-Sarpong et al. [41], and Kovacs [51] rightly call for an incremental yet expeditious approach to reform, and the old adage, "festina lente" (diligent haste) may

serve as a useful touchstone. While it is impossible to predict the future, we can proceed mindfully so as not to make those most likely to be impacted by change worse off.

### 4.2. Information Management Technologies

In a controlled construction manufacturing environment of the future, information management will determine productivity, efficiency, quality, and the safety of projects. It will also facilitate the incorporation of diversified perspectives and expertise. Significant articles, which are relevant to construction information management, were selected from the 289 articles identified in the research method section of the present study and analyzed. In Figure 5, the keywords co-occurrence analysis results using VOSviewer were implemented based on the selected articles. It shows the occurrences and average public years of the keywords and the networks between them. The most frequently mentioned technologies in the sample are BIM, IoT, DT, BCT, AR, VR, CPS, and AI. According to the publication year analysis, which is implemented as the color differences, the research tendency could be learned. The papers are from very recent years, from 2018 to 2022, and the figure shows that the research about BIM and AR are more aggregated in the year 2018, IoT, CPS, and BCT occur most frequently in the middle years, and DT and AI are mentioned most in the most recent years. Based on the analysis of the keyword, a more particular analysis of the key information management technologies likely to drive the future of construction is provided. These include BIM, DT, CPS, and semantic and ontology-based data management.

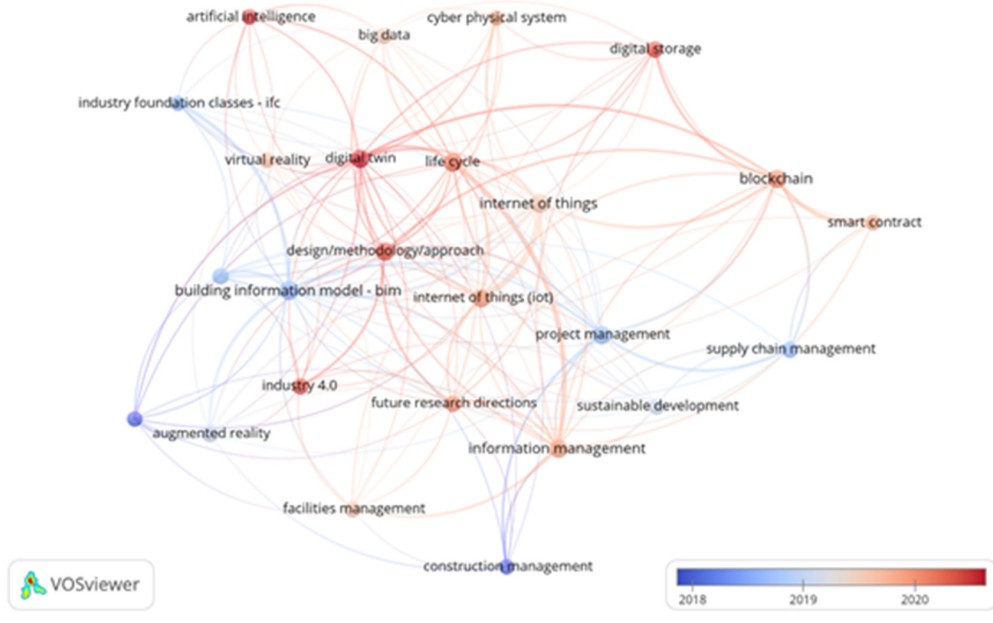

**Figure 5.** Overlay visualization of keywords occurrences and average publication years.

### 4.3. BIM and Its Integration with Information Analytic Technologies

BIM has been applied rapidly in the construction industry for the past 10 years [52]. The future research direction for construction information management is in the shift toward the integration of multiple technologies. A review of the future trends of BIM and its application, along with other information analytic technologies (e.g., geographic information systems (GIS), IoT, AR, and BCT are investigated.

In Table 3, the integration research about BIM and other technologies is concluded from the literature database settled in the article retrieval process. In this table, the importance of BIM in the past years could be proved as all the articles with combination applications are based on BIM. The increase in other technologies shows the research trends in the future, which is the integration of different technologies for information management of the construction industry. In Table 4, eight representative articles about the combination study

of BIM and other technologies are listed to illustrate the state-of-the-art current research status and future trends.

**Table 3.** List of papers referring to the integration of technologies in the FOCIT literature.

| | BIM | GIS | IoT | ML | AR | Semantic Web | BCT | Reality Capture | DT |
|---|---|---|---|---|---|---|---|---|---|
| Deng et al. [52] | * | | * | | | | | | * |
| Wong et al. [53] | * | * | * | | | | | * | |
| Rausch et al. [54] | * | | | | | | | * | * |
| Sijtsema et al. [12] | * | | | * | | | | * | * |
| Malagnino et al. [55] | * | | * | | | | | | |
| Gheisari et al. [56] | * | | | | * | | | | |
| Deng et al. [57] | * | * | | | | | | | |
| Dave et al. [58] | * | | * | | | * | | | |
| Das et al. [59] | * | | | | | * | * | | |
| Chen et al. [60] | * | | | | * | | | | |
| Chen et al. [61] | * | | * | | * | | | | |
| Williams et al. [62] | * | | | | * | | | | |
| Wang et al. [63] | * | * | | | | * | | | |
| Nawari et al. [64] | * | * | | | | | * | | |
| Khan et al. [22] | * | | | | * | | | | |
| He et al. [65] | * | | | | | | | | * |
| Darko et al. [16] | * | * | * | | * | | | * | |
| Alizadehsalehi et al. [66] | * | | | | * | | | | |
| Niu et al. [67] | * | | * | | | * | | | |
| Pauwels et al. [68] | * | * | | | | * | | | |

Note: The symbol * denotes the use of technology.

**Table 4.** Selected publications about the integration of BIM, GIS, and other analytic technologies.

| Literature Content | Publish Year | Statements | Prospects |
|---|---|---|---|
| BIM and IoT integration for facility management (FM) [69] | 2021 | BIM and IoT integration research are still at the early stage, as the works stay at the conceptual level. | BIM: the interoperability of data needs to be improved for FM; the industry foundation class (IFC) open standards need to be reviewed for the information demand of FM |
| BIM and IoT devices integration [70] | 2019 | The real-time data from IoT are connected to BIM models and the research about integration of BIM and IoT in the initial stage. The methods that have already been used are focused on BIM application programming interface (API) and relational database, query language, semantic web technologies, and hybrid approach. | Future research directions are suggested as service-oriented architecture patterns (SOA), web services-based strategies, standards establishment, cloud computing etc. |
| BIM, GIS, and Web integration [71] | 2021 | The integration research of BIM, GIS, and Web is in the tendency to grow especially after 2016. | Future research gaps are integration interoperability solutions, standardization, model processing, data exchange etc. |
| BIM and AR [72] | 2020 | The methods adopted in data capture for building site construction are fiducial markers, GIS, GPS, laser scanning, and photogrammetry. The integration of BIM and AR would enhance the visualization of the site and improve the information process for construction management. | It is recommended that the AR impacts on the quality, execution speed, loss reduction, and production increase of BIM-based projects are investigated. The validation of the integration model of AR and BIM needs to be implemented. |
| BIM and Image-based technologies [73] | 2017 | Image-based technologies in data capture, object recognition, and as-is BIM construction are reviewed. | The challenges could be decreased cost for data capture, improved efficiency for data management, pre-designed methods for object recognition, and full automation for as-is BIM construction. |
| BIM and BCT [74] | 2019 | The applications of blockchain in AEC industry and its incorporation with BIM are investigated. The distributed ledger technology (DLT) also improved BIM workflow on network security and data management, tracing, and ownership. | The hyperledger fabric (HLF) applications for enhancing automated code compliances in BIM workflow is the future prospect. |
| BIM and GIS: IFC geometry transformation [75] | 2019 | To realize efficient data exchange for the integration of BIM and GIS, this work enhances the open-source approach (E-OSA), by developing an automatic multipitch generation (E-AMG) algorithm. | The E-OSA enhanced by E-AMG still requires human intervention; it should be improved in the future. |
| BIM and machine learning integration [76] | 2021 | To improve information exchange for AEC projects and leverage data interpretation, the work proposed a system for property valuation. An integration method of BIM and machine learning is used, implementing database interpretation, IFC information extraction, and automated valuation model (AVMs). | The authors suggested infusion technology of BIM and other digital technologies like IoT, DT, BCT, cloud computing, machine learning and so on could be used for property valuation and the AEC industry. |

## 4.4. Digital Twin (DT) and Cyber-Physical System (CPS)

The DT paradigm is a more recent, integrated approach to micro (project level) and macro (urban level) modeling compared to BIM. As an ideal future city and construction

model, DT enables real-time web integration and intelligence, which could be applied to the whole life cycle of construction projects. It integrates the physical world with a virtual platform to control the construction process and environmental monitoring [52]. To that end, information management will evolve from the IFC format to more open linking building data to ensure "the right data is available at the right time" [18]. Several technologies are also required to realize the DT paradigm, especially considering the challenge of real-time processing between the physical and virtual phases. Sepehr et al. [77] emphasized the crucial significance of technology integration of BIM, XR, and DRX for construction progress monitoring based on DT. Sepasgozar [14] implemented the integration utilization of VR, AR, IoT, and DT, in education use for students to acquire knowledge of running construction machines and managing the construction process online. Ozturk G [78] suggested that capture technologies like sensors, gauges, machines for measurements, lasers, and vision facilities could seize the real experience data from the physical world. Rausch et al. [54] proposed geometric DT for offsite construction, implementing 3D scanning and a scan-to-BIM approach.

CPS is a broader concept to enhance the information models like BIM; it was developed more than 15 years ago and advocates for the interaction of cyber and physical elements [79]. DT provides a penitential realization of CPSs, on the scale of monitoring, simulation, optimization, and prediction [80]. The real-time sensors contribute to information communication for CPS, like IoT [81]. Investigations were done with the application of installation control for prefabricated modules, risk control of blind hoists, safety management, and so on [82–84].

*4.5. Semantic and Ontology-Based Data Management*

Semantic web and ontology-based data management will be vital for interoperability and information exchange in a construction manufacturing environment of the future. Its limited but growing use in construction data management is reflected in its position as a very special research hotspot in the literature. It is noticed that the semantic web technologies application in the construction industry is increasing. The use of the semantic web and ontology is typically treated as complementary to BIM and DT. Pauwels et al. [68] emphasized that the three main uses of semantic technologies are the interoperability improvement for information exchange among diverse tools and disciplines, the connection of obtained information to different domains, and the establishment of the logical basis in this domain. To draw a basic knowledge map of this concept, an independent literature investigation was executed. Among the literature corpus selected in this study, when the search items with "ontology", "construction industry", and "future" were examined, 30 papers were relevant to this sub-topic. The word frequency analysis of the most used 20 words in those papers is shown in Figure 6. It is proved that the ontology used in the construction industry is related to knowledge, information, semantics, HTTP, and IFC.

The OWL is in the full name of W3C Web Ontology Language. It is a semantic web language that aims to describe complex knowledge, groups, and relations of things [66]. This kind of language is designed as a computational logic-based language such that computer programs can exploit knowledge expressed in OWL. OWL is part of the W3C's semantic web technology stack, and OWL documents are known as ontologies. The ifcOWL ontology proposed by building SMART is a good example of OWL use based on IFC. As an open source and strict standard for the entire AEC industry, the ifcOWL improves information sharing and integration during the whole lifecycle of construction projects. When the construction industry faces a diversity of domains and disciplines of the whole building lifecycle of construction projects, the linked building data (LBD) is proposed by the LBD community group. LBD refers to using ontology or semantic web technologies for building data in the form of RDF graphs [85]. It emphasizes the interoperability of information sharing among stakeholders when using software through the internet and makes it easier to integrate into building elements without increasing the complexity of data querying [86].

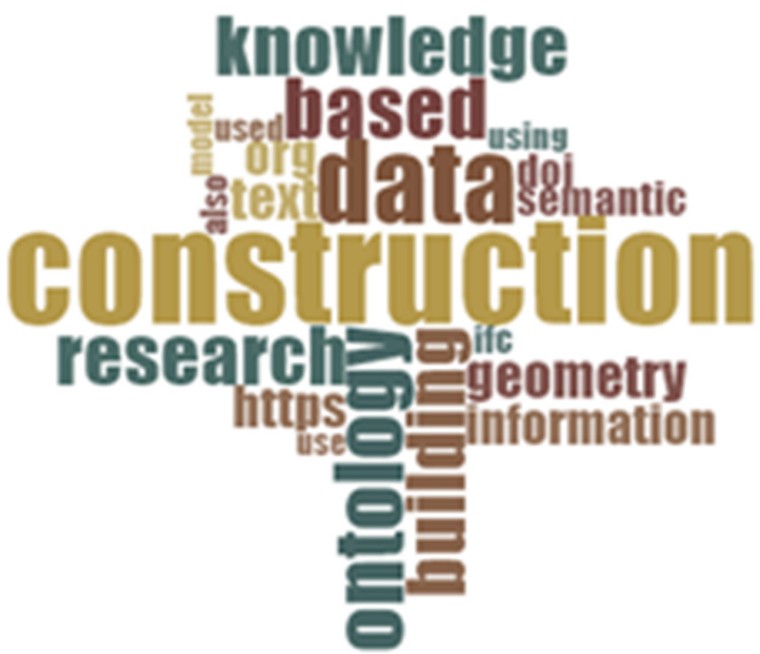

**Figure 6.** Visualization of keywords occurrences of construction related ontology use.

From AutoCAD to BIM, then to DT, there is no doubt that the construction industry has been embracing the digital generation for a long time and seeking further development with multiple technologies involved in the following years. In this prospect, data management from simulation and sensors faces problems of accuracy, storage, and safety, which could be solved based on semantic models [18]. The semantic and ontology-enhanced data management also grounds decision-making and facility management. It is also worth noting that semantics and ontology are the fundamental concepts for the DT and CPS, which could provide dynamic data flows at different scales [18,87].

### 4.6. Life Cycle Information Management and Sustainability Concern

From another perspective, various technologies will be adopted for information management at different stages for construction manufacturing projects, and some of them also contribute to the whole construction life cycle. The most used and recommended technologies for information management at each stage are listed in Figure 7, respectively.

The sustainability concern for information modeling occupies important status in the future of construction [88,89]. Construction and demolition waste (CDW) is thus a very important topic as the proper information system establishment for reducing CDW could decrease environmental damage [90,91]. Green BIM is expected to be used in the future, especially for high-rise buildings. Wang et el. [63] investigated BIM and GIS integration in sustainable built environments. Saieg et al. [92] presented a literature review about the integration of BIM, Lean thinking of construction, and sustainability, especially in decision-making. The results show that the interaction of these fields provides efficiency improvements in the future by reducing economic and environmental impacts [92]. Nevertheless, the attention to sustainability shows the researchers' determination to undertake social responsibilities. All the adopted, implemented, and developed technologies serve the goal of a sustainable life cycle of construction.

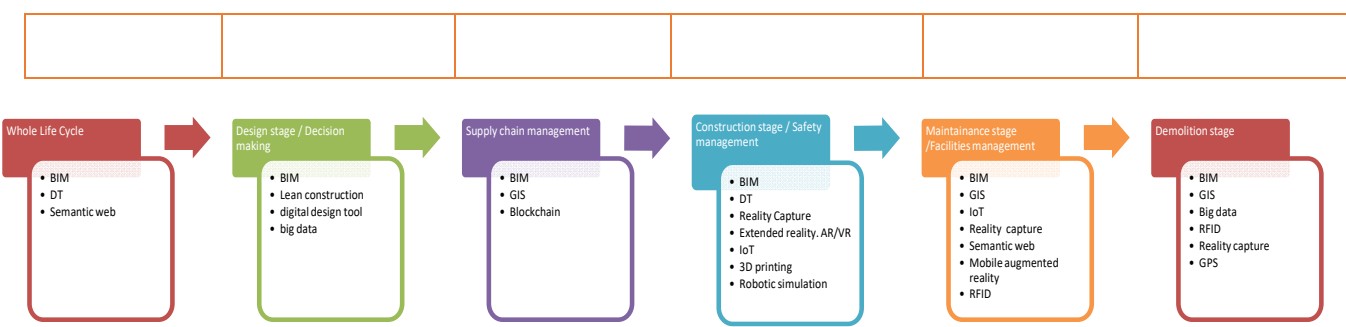

**Figure 7.** Information management tools at different construction stages. Note: Different stages—whole life cycle [18,52,65,93,94], design stage [10,88,92,95–97], supply chain management [32,57,98], construction stage [61,65,77], maintenance stage [52,53,56,78,85,99,100], and demolition stage [91,101,102].

## 5. Visualization Techniques

### 5.1. Current State of the Art

Visualization techniques will be an indispensable part of construction manufacturing, and the technology has been evolving dramatically over the last years, expanding to different sectors and audiences for applications in the AEC industry. Among the most popular technologies are VR for immersive simulations, AR for superimposed objects, and MR, which is a combination of the previous ones [103]. These are integrated with data collection techniques to enhance their realism and accuracy, promoting the development of prototypes across the construction sector. These approaches have involved areas such as safety management, data acquisition, and improved visualization of scanned objects, applying them for enhanced training modules and academic learning. In addition, the integration of technologies has played a significant role in new research, using them to provide better data processing, automation of processes, monitoring of assets, assemblies' recreation, and image tracking. Some of these projects aim to resolve long-lasting risks associated with the industry, such as site injuries [66,104], inspection and maintenance [60], fire safety awareness [61], and overall accident reduction [105]. Moreover, other researchers have focused more on training implementation, such as construction assemblies tracking [106] and educational training through digital technologies [10].

A major focused visualization technology that has been progressing over the last decade is the enhancement of data acquisition and representation of these digitalized objects, optimizing them to achieve a more meaningful and useful representation of data. Different fields were explored regarding data acquisition techniques; the role of cameras, laser scanners, and drones for progress monitoring and georeferencing [77]; and software involving BIM and DT for its optimization [77]. Uses of unmanned aerial vehicles (UAVs) for georeferenced photogrammetry [107], holography for enhanced 3D representations [108], and BIM and image-based technologies for operations and management (O&M) [73] are some examples of data collection alternatives. In addition, the characterization of a green façade with laser scanners [109], offsite data capturing of assemblies with DTs [54], facilities management with AR/MR approaches [56], and behavioral analysis of occupants with VR [110] have represented examples of data extraction and interpretation.

Moreover, integrating different technologies has rapidly evolved into more complex applications, using their strengths and weaknesses to complement each other. Proposals such as the integration of GIS, BIM, and XR are explored for designer/engineer roles in the whole lifecycle project, supporting uses around decision-making, design, training, and education [56,66]. Additional explorations are also proposed regarding cloud points and software processing optimization, such as DTs, AI, IoT, ML, and cognitive computing processes [77,111]. In conjunction with the BIM or DTs approach, AR and VR were implemented as a solution for site inspection, maintenance, installation, and training in the professional and academic areas [25,72]. Also, the possibilities of cloud computing and

the 5G network were explored with XR implementations, presenting alternatives for close real-time processing and visualization [11,81].

### 5.2. The Role of Immersive Technologies

Immersive applications will revolutionize construction manufacturing, creating multiple benefits but also risks associated with their adoption. Researchers have carried out SWOT analyses for immersive technologies, identifying these major impacts on stakeholders, design analysis, construction planning, facilities management, and education/training [22]. Strengths such as better perception and productivity are also mentioned, highlighting weaknesses due to hardware limitations and field usability of immersive headsets [112]. However, the authors also alluded to the opportunities that VR and AR are capable of, using them as tools for better communication and integration of activities. These technologies are often associated with enhanced upskilling of personnel, having the potential of reducing reworks, increasing safety perception, lowering the cost of labor, and improving project deadline deliverables by complementing simulations with experts' knowledge of their field [106]. With these digital environments, immersive technologies can produce new ways of interaction, incenting multiuser collaborations, and interactive equipment handling for complete training before going to the site [105]. Similar concepts were implemented by Sepasgozar [10] in their work, showcasing examples of an AR excavator and a VR tunnel boring machine module for equipment teaching in the education sector. Therefore, with all these applications, it is possible to enhance multiple activities' overall construction manufacturing process, decreasing the quality dependence on the worker by supporting their jobs with immersive technologies [25].

Nevertheless, multiple challenges need to be addressed to effectively implement VR and AR in the AEC industry life cycle. Increased development costs and low financial justifications are among the recurrent limitations that companies and researchers perceived. This needs better financial impact analyses to suit tight project deadlines and specifications [103,104]. Restrictions regarding software and hardware are also present, having limitations on the applications that can be produced by the low data processing of virtual environments and restricted storage [66]. This creates problems associated with the accuracy of collected data because of inconsistencies with the cloud points gathered from UAVs [107]. Bello et al. [11] suggested the use of cloud processing to overcome these limitations; however, in the light of proposing this, other issues arise, such as latency, data availability, band connectivity, security, and expensive charges for cloud usage. This leads to the importance of integration, where all the downsides of VR and AR implementation need to be addressed to overcome the resistance to change, merging these new ways of doing things with already established traditional methods [106,111]. This resistance is attributed to the slow adoption of new practices by the AEC sector related to experience, age, social influence, and satisfaction affecting the perception of potential users with immersive technologies [14]. Other challenges involve the existing knowledge gap in managing these technologies, companies' low availability of resources, and inconsistent standard practices that hinder their adoption [112]. These barriers for visual technologies must be addressed while developing practical applications relevant to the industry, justifying how they contribute towards the overall project lifecycle rather than only focusing on some areas of a macro process.

### 5.3. Visualization Techniques Prediction

Multiple studies suggest what can be future trends or solutions for visualization applications, giving an insight into how new approaches should be carried out for meaningful results in a hard-to-change industry such as the AEC sector. AI-assisted VR systems will improve substantially to replicate the randomized behavior of humans in a cyber environment, creating applications where multiple users can interact in real-time with close to no delay [113]. Virtual activities will be able to accurately represent the real procedures for any construction activity, feeding them with the empirical experience of professionals to create an effective learning module [60]. The standardization of VR and AR practices will

be developed, establishing metrics to assess their effectiveness and manage their implementation in the project life cycle [25,62]. This involves the creation of protocols to test the quality of simulations, reviewing and optimizing them to increase processing data and thus reducing hardware expenses with smarter AI algorithms [114,115]. Future immersive technologies applications will integrate multiple disruptive technologies with current traditional methods, complementing each other with their advantages rather than annihilating the other [104,106]. In this sense, the integration will be the pillar to justify technology adoption in the AEC industry, using DTs, GIS, BIM, data collection, cloud processing, IoT, ML, and AI for feasible applications [12]. Financial analysis will be essential to justify immersive technologies adoption, verifying what is achievable for each application in a specific timeframe, just as real projects are managed [103]. All these aspects are paramount to achieving the vision researchers have about the future of immersive technologies, putting an emphasis on the cultural background of stakeholders to suit their individual needs for an effective adoption in the AEC industry [116].

As seen in multiple publications, the most common limitation for visualization techniques such as AR, VR, MR, and XR is related to the restrictions of hardware and software. Therefore, multiple predictions of visualization techniques for the future regard improvements in these two areas. Future head-mounted display (HMDs) devices will have better accuracy, precision, and positioning, using prototypes of Microsoft HoloLens and Smart Glasses [72,103]. Cloud storage and processing will become the standard for data management, using personal servers and 5G to 6G technologies to reduce the latency of information and improve band connectivity [11,81]. These trends will enable the use of powerful algorithms, supporting them with AI and ML to greatly reduce data processing times and noise interference [77]. New possibilities will be created with better software and hardware implementation, moving immersive technologies towards the use of 3D volumetric holograms and real-time editions of BIM models [108]. These new devices will have improved battery life and better ergonomic design, making them suitable for extended sessions and rough conditions of construction sites [22].

*5.4. Immersive Technologies in the Future*

Immersive technologies have been gaining increased recognition over the last decades, experiencing higher levels of research development and investments by stakeholders involved in the AEC sector. Practicians have recognized its value and benefits; nevertheless, there is no clarity on how much immersive technologies are worth now or in the future. Research market analyses hint that AR, VR, and MR value will be above billions of USD between 2022 and 2023, all of them predicting an increase in these technologies for the next 10 years. However, the level of confidentiality these companies have with their assumptions and sources makes it very difficult to conclude what estimations are the most accurate ones. Therefore, in Figure 8, the publicly disclosed data by these companies is presented as the Compound Annual Growth Rate (CAGR), classifying the information into the categories of VR, MR, AR/VR/MR, AR/VR, and AR. These growth rates were obtained based on the relative information each company predicted as the initial and forecasted value of immersive technologies, allocating the percentages between the analyzed timeframe of each insight.

As showcased in Figure 8, the growth estimations of immersive technologies are expected to be between 18.01% and 44.80% for VR, between 38.15% and 83.30% for AR, 41.80% for MR, between 34.70% and 73.70% for AR/VR, and between 66.84% and 113.05% for AR/VR/MR. These data reflect a higher expectation for immersive technologies when combined, suggesting the integration of practices to achieve meaningful results, as highlighted in the previous section. In addition, when analyzed individually, AR is the most promising technology of all; this is in line with the perception that Hamzeh et al. [108] highlighted about the increased interest of the AEC industry in AR applications. The versatility of superimposing virtual elements into reality is among the most anticipated

benefits practitioners expect of immersive applications, being paramount to the focus of new research to integrate it into the current and future head-mounted displays effectively.

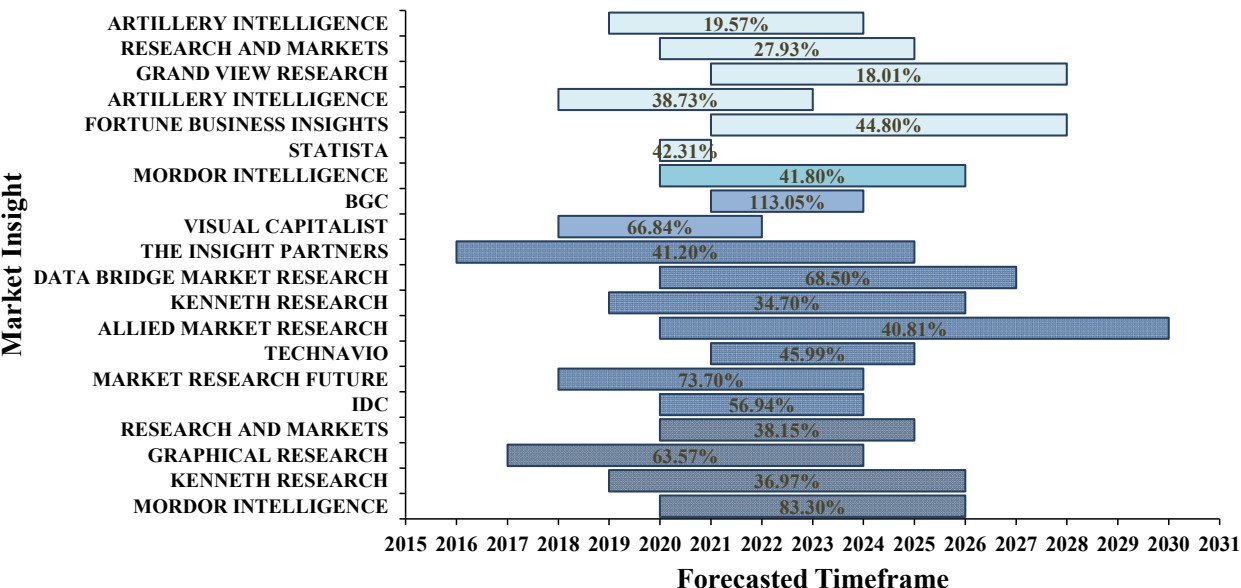

**Figure 8.** Estimated compound annual growth rate for immersive technologies market value (source—[117–119]).

The integration of not only immersive technologies but also the inclusion of new data collection techniques, machines, AI, storage, and processing will bring new benefits to the future of construction with no apparent limits. The use of remote-control rovers (a) and UAVs (b) for laser scanning and photogrammetry will become a common practice on construction sites, as seen in Figure 9, taking advantage of cloud processing and storage to optimize and visualize the information almost in real time.

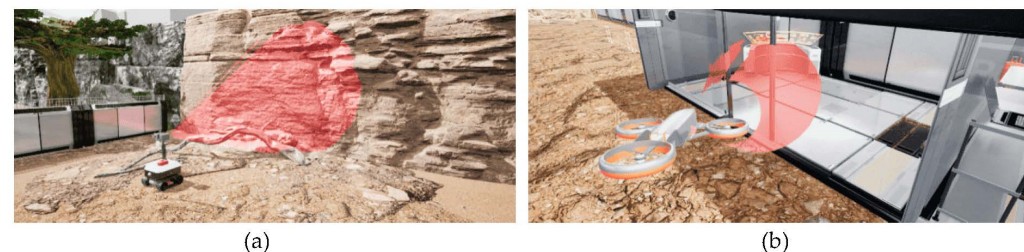

(a)                                                                          (b)

**Figure 9.** Future data collection techniques (**a**) Laser scanner rovers, (**b**) survey UAV (source: Author).

Information captured by these machines will be instantly transmitted to the main servers through 5G or 6G connections, integrating it with immersive devices for display. Although useful, the use of UAVs should have safety and privacy guidelines for personal life protection as well as for society. Regarding this, many jurisdictions, such as the US, the European Union, and Japan, have specific guidelines and policy recommendations, and other developing and developed countries need to develop their own regulatory frameworks and policies. A state of art comparison of UAV policies and regulations can be looked upon by Lee et al. [34], wherein the study comparatively analyzed the safety and privacy regulations of UAVs in different jurisdictions.

AR and MR will be used as techniques to interact with the collected data, reviewing, in real-time, the surveyed information in a command center located in the city, as presented in Figure 10. From this location, managers can remotely control and supervise the construction sites, gathering real-time statistics of site conditions to display as dashboards. With time, immersive technologies will progress until 3D holographic displays become the standard for data visualization, facilitating coordination and decision-making among stakeholders.

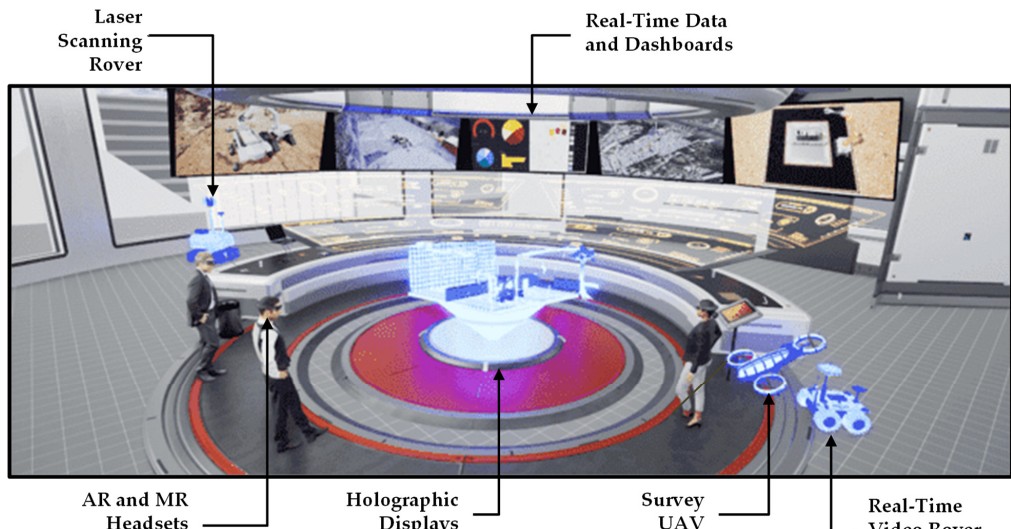

**Figure 10.** AR and MR utilization in the future of construction (source—Author).

VR will move forwards to replicate more realistic scenarios, creating diverse training modules and teaching modules comparable to actual activities. Environmental factors, safety management, handling operations, variable site conditions, and user engagement will be essential elements for feasible prototypes relevant to the industry, as represented in Figure 11. Multiuser collaboration and avatar interaction will be crucial components for 2030 applications, delivering new alternatives to enhance personnel upskilling and engaging learning modules.

The future of VR will be oriented to enhance learning and interactive spaces, following the ideals of the Metaverse as Facebook presented in Connect 2021. Social interaction and business work will be fomented with avatars and immersive devices, implementing them to create a universe of interconnected virtual environments [120]. This will give users the flexibility to do things only possible in simulations, enhancing the communication and processes of the AEC industry.

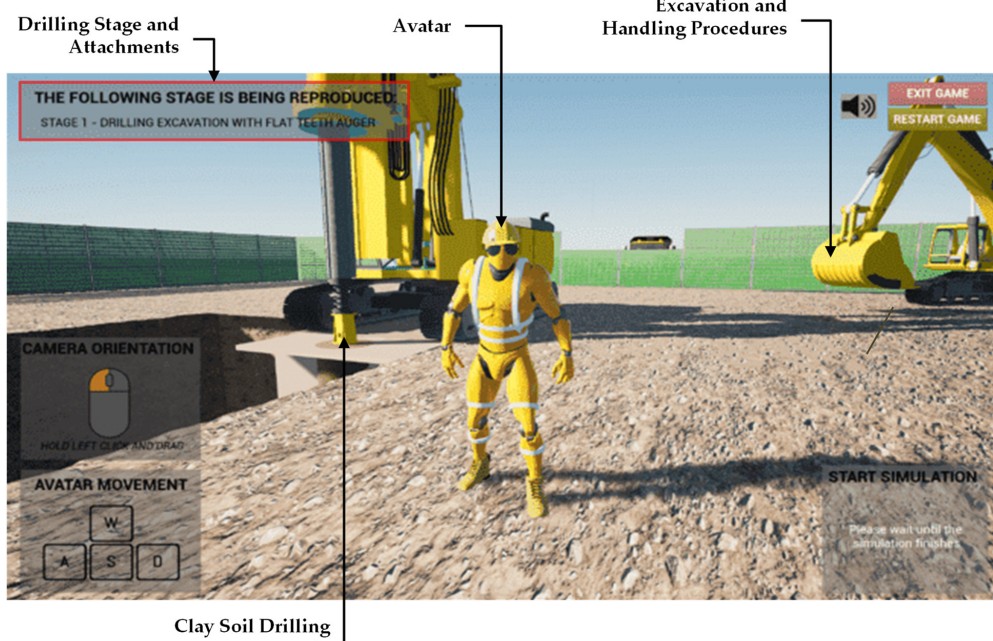

**Figure 11.** VR pile training module (source—Author).

## 6. Discussion on the Roadmap for the Future

The future of the AEC industry depends on disruptive technologies due to rapid growth in digitization and digitalization of the industry [2]. The experience of working during COVID-19 motivated the innovators to investigate and offer new solutions enabling the practitioners to work remotely in a productive manner [121]. Throughout the lifecycle of a project (design and planning, construction, operation, and management), the advent of proposed disruptive technologies will push the boundaries of the AEC industry to develop novel solutions and business ideas for each phase of the lifecycle. Whether talking about streamlining the design process, automation of construction tasks, assessing the workers for dangerous tasks, smooth operation of a project during the lifecycle, and environmentally friendly demolition processes, the disruptive technologies will be an asset to any stakeholder relevant to the AEC industry. Following this backdrop, 11 disruptive technologies are discussed as follows that will revolutionize the AEC industry in the next 10 years.

### 6.1. 4D Printing

The technology known as 4D printing is advanced 3D printing (3DP) with the flexibility of property changes in objects with changing environments. The additional dimension represents the time that transforms the 3D-printed object or asset or becomes another structure because of external resources such as light, heat, magnetic field, temperature, electricity, and a few other provocations [122]. Using 3D printing as a backdrop, it can perform automatically, proficiently, more flexibly, and has strong mechanical properties, among others [123,124]. These factors facilitate in reducing construction time, generating different geometry within the object with extraordinary strength and resistance to corrosion and high temperatures. In addition, the materials are environmentally friendly and organic, which adds to the sustainable aspects of 3D printing. However, 4D printing is similar to 3D printing in terms of digital printing of the objects; it differs in using smart materials with unique thermomechanical properties that can change their shape and are based on programmable technology [125]. The programmable technology applies the self-assembly process to re-envision the process of building and production. Although 4D printing has been applied in various fields like manufacturing and medicine, its utilization in construction is still limited or in the experimental stage. However, a recent study by Hwang Yi [126] implemented 4D printing to develop a prototype of a parametric smart façade that can acquire self-shaping skin and could be significant in adaptive building design and construction. Nonetheless, 4D printing is one of the disruptive technologies in the AEC industry that will create innovative design and construction ideas and implementations in the future.

### 6.2. Augmented, Virtual, and Mixed Reality

Immersive technologies will revolutionize how information is shared, displayed, processed, and modified for the year 2030 in the AEC industry. AI and multiuser applications will be developed to create new ways of interaction, enhancing training and learning processes with the aid of VR [113,127]. Cloud storage and processing will be the main vehicles to overcome hardware and software limitations, having benefits such as scalability of resources in a case-by-case situation [11]. Future immersive technologies will be integrated with other applications to create meaningful results, such as more realistic scenarios through cloud points, lower latency with 5G signals, data processing automation with machine learning, and site analyses with DTs [12,73]. Head-mounted displays will be more ergonomic with increased battery life and robustness, suitable for outdoor site usage [22]. AR and VR will be integrated with traditional standards to overcome change resistance, promoting collaboration between practices instead of a complete replacement [104,106]. New immersive applications will adopt metrics and standards to identify their benefits across project lifecycles, ensuring results follow minimum quality standards for effective applications in a construction environment [22,103]. Finally, empirical experience and professional expertise will be fundamental inputs for realistic VR prototype implementations,

integrating the information with high-detail cloud points obtained from photogrammetry and laser scanning [60,103].

Immersive technologies are forecasted to grow for the next 10 years, getting more recognition from stakeholders about their future potential. Its market value presents fluctuations with unclear assumptions by market research companies; however, all of them coincide that these technologies will continue growing even after the impacts of COVID-19. There is a higher expectation for mixed approaches between AR, VR, and MR, highlighting that augmented reality expectations are higher than the other two when analyzed individually [106]. The contribution of new technologies, such as remote-control machines and survey-collecting systems, will transform how data is managed, implementing cloud elements to visualize it in real-time with the aid of 5G signals [128]. AR and MR will be implemented for surveyed data interaction, reviewing the site status from a centralized control center in real-time. Statistics and site measurements are going to be displayed through head-mounted display devices, and they will continue to progress until 3D holograms become the standard practice for visualization, coordination, and decision-making. As VR refers, its future applications will focus on virtual multiuser collaborations, focusing on training and teaching modules that accurately represent the variable conditions of a real construction site [125]. Avatars will be used to represent users in cyberspace, connecting this vision with the Metaverse as defined by the Facebook vision of the future [120].

### 6.3. Cloud XR

XR stands for extended reality, which encompasses virtual reality, augmented reality, and mixed reality. All these realities come under the umbrella name of extended reality and have different significances in the AEC industry [66]. Visualization of the design process, real-time animation, training of site workers, estimating hazards in dangerous scenarios, safety training, schedule control, optimization of site layout, collaboration environment simulation, progress monitoring, and control of site activities are some of the benefits achieved through XR utilization in AEC industry [22]. However, a few issues, such as accuracy and tracking process enhanced positioning and mapping, and multiple sensory integrations exist, which are due to many factors, one among them being because of low network and connectivity. The emerging 5G network integrated with edge cloud computing technologies presents an evolutionary path to XR's based on the cloud [11,129]. Cloud XR delivers the XR content streaming on a remote cloud server with any open XR application. Through Cloud XR, collaborative value monitoring is achieved, which could concretize the information handling potential from the different computer devices, thus delivering receptive shared responses in addition to real-time acuity. This process will also lend a hand in facilitating the digital twin process for enhanced monitoring of various tasks and procedures. A recent study by Alizadehsalehi and Yitmen [77] utilized XR techniques and reality capture to develop a generic framework of digital twin-based monitoring of construction processes. The research provides steps to combine XR, reality capture, and digital twin to create, capture, generate, analyze, manage, and visualize information in real-time. Cloud XR's future lies in better integration with BIM to provide assessments regarding the design process, safety, progress, and construction monitoring, among others. A holistic framework architecture of Cloud XR is required to provide details on how frequently images or videos from cloud servers can boost the process of XR services to configure prompt simulations and instinctive information [125]. Further, mutuality among different tasks is also needed to deliver unambiguous information and data.

### 6.4. Metaverse for AEC and VDC Professional (M-AEC)

The Metaverse, derived from a 1992 sci-fi novel, is deemed a successor to the internet coined by Neal Stephenson [130]. In a nutshell, it is a digital world where anything imagined can exist; it will be a connected source with time and will have the capabilities of augmenting the senses of humans, such as sight, sound, and touch. Until today, the Metaverse concept is still in its evolving stage, with no concrete definition identified.

However, a recent study [131] proposed a framework for the Metaverse with seven layers, including infrastructure, human interface, decentralization, spatial computing, creator economy, discovery, and experience. The related technologies for supporting the Metaverse are extended realities (VR, AR, and MR), artificial intelligence, computer vision, edge and cloud computing, future mobile networks (preferably 6G), blockchain, and non-fungible token (NFTs) [130]. The medium to enter the Metaverse will be extended realities (VR, AR, MR). Implementation of tasks in the Metaverse will be significantly authoritative for AEC and VDC professionals to muddle through clients' and stakeholders' expectations making them immersed to have telepresence and interaction with the projects. The likes of remote collaboration, prototyping, BIM, and VDC coordination, finishing, and presentations will be streamlined in the Metaverse. A few use cases or prototypes have been developed, such as the concept of "charter city" and "prospera".

Amidst benefits, the challenge with developing the Metaverse revolves around users and their avatar identity, content creation, social acceptability among the masses, presence security, accountability and trust, privacy concerns, and the virtual economy involved [130]. A major challenge for the AEC/VDC industry will be the development of a legal framework for the verifiability of the agreements in the virtual world and dispute resolution practices. In a recent study, Wang et al. [130] proposed three aspects that will govern the Metaverse's suitability in the AEC industry, namely "intelligent combination of smart technologies", "intelligent recognition reasoning and decision making based on knowledge graphs", and "multi-machine andhuman–machine collaboration". Nevertheless, the Metaverse will prove to be a new headquarter or a central hub to empower professionals to access their assets from anywhere in the world.

### 6.5. AIoT (AI-Integrated IoT)

Although there is no accepted definition of the Internet of things (IoT), it is regarded as a comprehensive, interconnected network of physical tools comprising drones or UAV's, sensors, radio frequency identification (RFID), extended realities (VR, AR, and MR), a global positioning system (GPS) and other sensing, communication, and actuating tools [129]. Integrating with other technologies, IoT revolutionizes the Internet itself and perhaps acts as a bridge to culminate real and virtual environments. Nevertheless, utilizing IoT delivers only static data; combining it with AI can provide dynamic data, information, and insights [132]. The construction industry is plagued with circumstances and issues that require future predictions and forecasts, which can be leveraged through amalgamating AI and IoT-powered devices. Therefore, AIoT can be called a novel term integrating AI and IoT to deliver enhanced data analytics, insights, and operations [125]. Although many studies implemented IoT-based sensing technologies in construction, AI integration can facilitate better decision-making and predictions for construction operations. AIoT, with other techniques, can be used in the construction industry for high-definition surveying, geolocation, data collection, and acquisition. The dynamic tracking of the construction progress, job site monitoring, health and safety management, and logistics and management are some of the attributes of using AIoT solutions in the construction industry [132]. While many studies utilized IoT, the upsurge of AIoT is still new in the construction industry due to a few issues of cybersecurity and edge computing, etc., as reported in previous studies. AIoT has the potential to analyze the vast amount of data generated by IoT-driven devices to produce valuable insights for the future. In addition, the combined effect of BIM and AIoT in light of fast networks will facilitate the actualization of smart cities and infrastructure within the construction industry in the future [69].

### 6.6. Autonomous Digital Twin

Regarding the AEC industry, DT is a virtual, digital, or synthetic representation of a built asset that paves the way for better simulation of the models and forecast predictions to achieve enhanced decision-making. Integrating three key elements, physical, virtual, and data entwined together, is significant to achieve dynamic mapping for a DT [14]. The

CPS of BIM, IoT, and other data mining techniques is used to inspect the information in the physical asset to transfer it to the virtual asset. Further, this information is utilized in simulation, prediction, and optimization in the virtual asset to deliver solutions for different problems and issues to be implemented in the physical asset creating a synchronized loop. The maturity level of the DTs can be categorized into four levels, namely "pre-digital twin", "informative digital twin", "performance digital twin", and finally, "autonomous digital twin" [18]. Most of the research and industry trends now revolve around the first three levels, which reflect real-time data of the physical asset to the digital asset and vice versa. This allows for performance measurement and monitoring of operational activities [77]. It must be autonomous in nature to maximize the gains from the digital twin, including AI techniques, machine learning, and deep learning models to process the data, enabling it to generate new knowledge and insights [78,82]. Continuous learning through the data and improvement over time will lead to decreased downtime, better energy optimization, and the inception of new business models to provide a true value of the digital twin [52,82]. The resultant insights will foster the different problems of the construction project, namely automated site progress, pre-detection of issues, knowledge of safety and health-related problems, optimization of logistics, and scheduling processes, among others. Due to the continuous rise of computing technologies, autonomous digital twins will become a reality sooner than later [82].

### 6.7. Automatic Guided Vehicles (AGV)

One of the serious challenges in the construction industry is manual operations and handling of assets which decreases the overall productivity and speed of the project. There is a need for efficient and effective solutions to this burgeoning problem in the construction industry. In addition to the digitalization of intangible assets, logistics also need an automated approach, as seen in a few other industries like manufacturing and maritime. In this regard, automatic guided vehicles (AGVs) are a technology to adopt in the construction industry for the horizontal transportation of assets without human intervention [133]. There is minimal utilization of AGVs in the construction industry in this era of automation. AGVs can integrate with IoT devices to deliver seamless flexibility to control them while sitting in a secured control center [133]. AGV system includes guidance, navigation, power, and communication systems that enable the vehicle to move around in a controlled manner and on a programmed track. The utilization of AGVs in the construction industry has economic, social, and technical advantages. Reducing factory logistics, minimizing waste, and saving time in transportation are economic advantages factors. Socially, factors like worker safety, reduced WMSDs, reduction in noise pollution, and decrease in carbon emissions are huge drivers to utilizing AGVs in the construction industry [134]. Finally, the technical benefits of using AGVs lie in automating traditional and offsite construction processes by integrating other digital technologies such as RFID, and GIS. Although AGVs have significant potential in the construction industry, future directions toward establishing a streamlined economic-social-technical application framework are required.

### 6.8. Exoskeletons

An Exoskeleton is a system used to augment a person's physical capabilities [135]. It is often confused with robotic devices; however, it differs in providing a wearable exosuit/super suit to workers in various industries. Exoskeletons provide a shell covering on the wearer's body to allow enhanced strength and endurance during arduous tasks [136]. Exoskeletons are based on active and passive measures and categorized as back-assisted, shoulder/arm-assisted, leg-assisted, and full-body assisted. With the integration of position sensors, actuator controls, and fast signal processing, the exoskeletons can assist workers in strenuous tasks such as heavy lifting, holding, bending, and squatting-related work [136]. Although exoskeletons have been around for some time, their applications in the construction industry are still in their infant stage. The reluctance is due to the industry's acceptability in terms of cost-benefit measures and other potential barriers related to the

health and safety of workers. Studies related to application metrics of the exoskeletons are required that can provide a detailed explanation of factors such as ease of use, comfortability, investment willingness, and implantation guidelines, among others [135]. A collective effort to provide benefits and barriers will streamline the understanding and acceptability of exoskeletons in the construction industry in the future.

*6.9. Construction Telematics and Neural Controlled Devices*

As the construction industry is plagued with tight margins and often occurring downtime, the facilitation of communication, sharing, and data analysis are of utmost importance to avoid disruptive effects on the overall workflow of the project. Construction telematics is about managing asset data, whether a piece of equipment or a complete building [137]. By incorporating CT, the exact location of assets can be tracked, job sites can be managed, asset utilization for maximum output can be retrieved, and equipment life cycle can be predicted [138]. All this can be performed and tracked while sitting at a project control center located near or away from the construction site. Through the utilization of CT, companies can always monitor their assets' location, significantly manage projects, and job sites, maximize the output of utilized assets, and monitor the equipment maintenance cycle, among many others [137,138]. In terms of assets, raw materials on the site, various vehicles on the road, equipment used on site, and safety compliance of laborers can be properly managed by the application of CT. There are many datasets that are generated at a construction site but not properly managed to deliver useful insights. The need of the hour in the construction industry is to develop software platforms that can manage all the data of on-road and off-road assets to deliver useful insights rather than speculations.

Although still in its conceptual stage, the neural controlled devices (NCDs) are abrain–computer interface that will allow controlling the construction devices remotely using neural facilitated digital humans [139]. Conventionally, it can be regarded ashuman–machine interaction and collaboration to streamline construction tasks and operations [140]. For instance, on-site equipment in the construction industry will be mutually connected with the human brain to facilitate the tasks, thus reducing collisions or any other danger [141].

## 7. Integrating Circular Economy (CE) with FOCIT

The circularity percentage in the global economy is currently 7.2% which leaves an enormous gap of around 93% globally [142]. The linear economy method of take-make-dispose is the culprit as around 100 billion tons of material is consumed annually, which is estimated to reach around 170–184 billion tons in the year 2050 if the businesses tend to run in a typical and traditional manner [143]. Considering this circularity gap, the construction industry is one of the tarnished villains in widening this gap [144]. This calls for integrating CE principles in the construction industry sooner rather than later as construction is widely practiced on the liner economy methods of materials production and consumption [145]. Integrating CE principles in the construction industry is necessary for climate resilience as construction is a resource-intensive sector producing the largest ecological and carbon footprint [28]. The construction industry's current processes, products, and business models are responsible for depleting an enormous amount of energy, virgin materials, greenhouse gases, landfill wastes, and non-biodegradable yields, among other potential threats [146].

The Ellen MacArthur Foundation highlights the utilization and implementation of circular design principles, circular business models, industrial disassembly processes, reverse logistics practices, and government policies, among others, as the key enablers for CE [145]. Furthermore, the integration of novel engineering processes along with digital technologies is deemed to play a significant role in the transition of CE principles from theory to reality in many industries [144]. Although previous attempts in the literature, discussions in the industry forums, and the government's role has talked about policies, design strategies, and other enablers of CE implementation, there is a lack of circular business models and novel engineering processes in the construction industry [29]. Due to

this, professionals in the construction industry have found it hard to execute the theoretical principles of CE in original construction projects [147].

Although the digital technologies in the FOCIT domain discussed in this paper are essential to CE accomplishment, the integration of both is still immature and needs further integrative application frameworks and business models [148]. The application of digital technologies can be fruitful at different stages of CE cycle resource strategies, namely Regenerate, Narrow, Slow, and Close the Loop resources [26].

## 8. Mapping the Technologies

This section provides the holistic framework for mapping the technologies into three categories of emerging, disruptive, and convergent technologies towards their implementation in the lifecycle phases of AEC projects. The current practice towards the utilization of these three technologies in the AEC field is fragmented and plagued with a lack of comprehensive and amalgamated approaches. The authors developed a mapping framework of technologies (Figure 12), which can deliver seamless communication of the different stages of a construction project. Figure 12 is self-explanatory and requires no further explanation. The industry is predicted to shift from using technologies to more efficient and effective technology utilization over a time span of almost 10 years. Other than traditional project management objectives of time, cost, and quality with measures of productivity and efficacy, there are stronger motivations to embrace new technologies. These motivations are the world's commitment to neutralizing carbon emissions and the need for remote working solutions due to the COVID-19 pandemic. These motivations may transform the construction industry in the near future. Table 5 summarizes key topics that should be investigated as potential solutions on the horizon of 2030s.

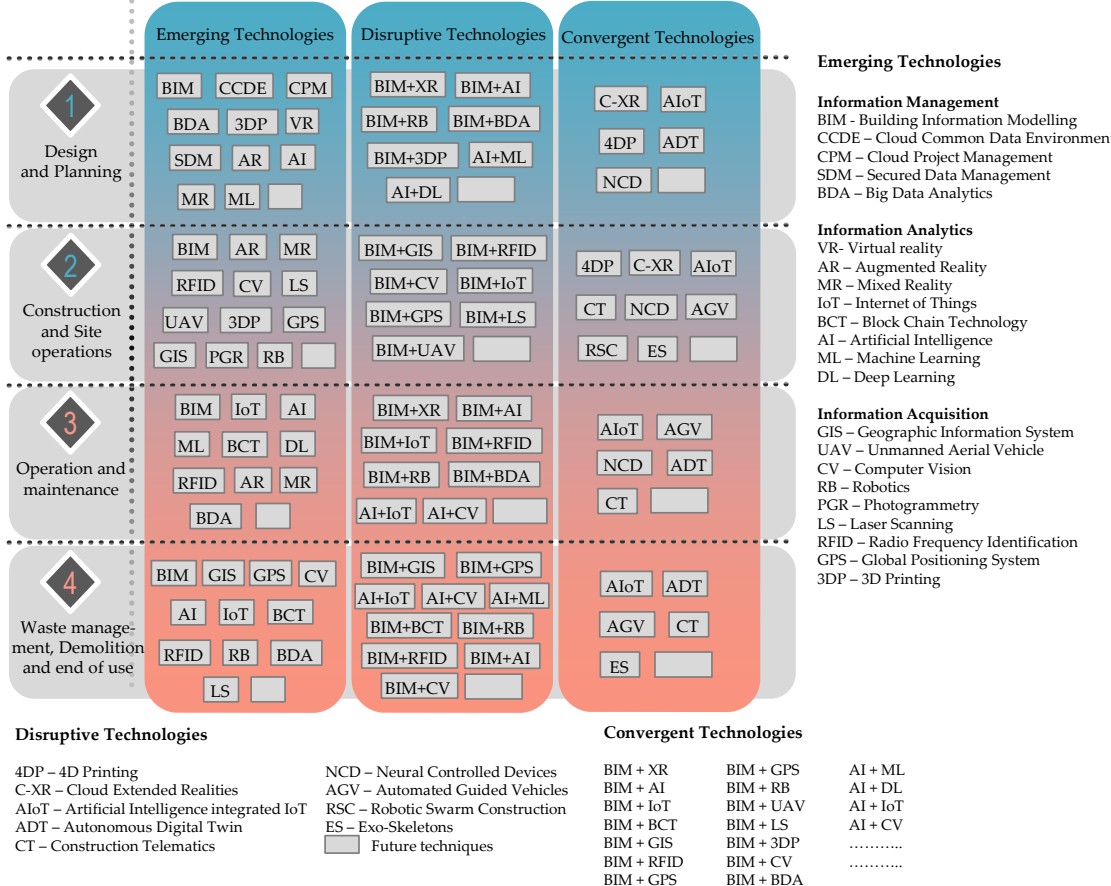

**Figure 12.** Mapping the technologies to their relevant stage of work.

**Table 5.** Suggestion for future studies based on a summary of potential solutions at the horizon of 2030s.

| Themes | Current Agenda, Emerging Concepts, and Technologies | Future Directions and Practices in 2030s |
|---|---|---|
| Sustainable construction and net-zero carbon emission | Paris Agreement on Climate Change; sustainable development goals (SDGs); awareness and policy development; case studies; circular economy | Use electric equipment for autonomous operation Embrace net-zero carbon emission Utilize AIOT-based supply chain systems to neutralize carbon Eliminate the waste by using 3D printing, modular off-site construction, and autonomous robots Lighter-weight, easy to install or use, higher strength per weight |
| Online and cloud technologies | Resilient during pandemic | Utilize remote control systems Users collaborate with robots |
| User interfaces and applications covering the entire life cycle from design to performance | Semi-autonomous excavators/bulldozers for some repetitive tasks, accurizing terrain data, and measuring productivity | Utilize standardization, repetitive design, and modularization to enhance robots' efficiency. Use of autonomous haulage systems and equipment |
| Platforms and controlling systems | Digital technologies are used for the design and Engineering, Construction, and Operation phases. | Integrate BIM/GIS and blockchain to share the models and information with all stakeholders in all the phases Use intelligent contracts to decrease disputes and enhance the efficiency of communication among stakeholders |
| Systems and dashboards | BIM and GIS are integrated into some projects. Interoperability and integration of various tools such as BIM, DT, blockchain, and the Metaverse. See more about the concept of interoperability [149] | Implement Open BIM and Open GIS. Extend collaboration in a cloud version. Develop connected BIM: e.g., BIM-GIS integrated with visualized dashboards that are easy to use by authorized stakeholders. BIM-GIS is a part of City Digital Twin and is used for performance optimization and impact assessment. Connect to sensors: BIM-GIS is connected to sensing technologies for the entire site, including heavy equipment and job-site tools. |
| Digital/physical integration systems and immersive technologies (VR, AR, and MR) | Measuring and connecting, mainly on-directional data exchange and simulation, basic asset digital twins | Predict and develop expert systems Bi-directional data exchange Connect to robots and enable remote operations Sensor fusion and data integrity Integration between multiple technologies, standardization, multi-user collaboration, real-time analyses, and the Metaverse |
| New materials, including nanomaterials driven by carbon fibers | | Lighter-weight, easy to install or use, higher strength per weight Eco-efficient and eco-effective materials drawing on concepts of Circular Economy developed by Ellen MacArthur foundation |
| Ontology and semantic web | The basic logic for data management as shared conceptualization and complementary to BIM and DT | Serving for interoperability, linking data, and logical inference |
| United Nations SDG U.S. Innovation and Competition Act (2021) Build Back Better Bill Industrial Internet of Things (IIoT) Free-market innovation R&D Tax incentives Green financing | Industry 4.0 Smart construction Digital transformation Horizontal integration | Made in China 2025 Fourth Industrial Revolution Construction manufacturing Cyber-physical systems Vertical integration Net zero Circular economy |

Table 5 provides information for future directions in line with two main questions. First, how digital technologies accelerate addressing climate change challenges by decreasing carbon emissions, waste, or other strategies. The second question refers to how technology enables businesses to continue working remotely during pandemics such as COVID-19. The third question is how to develop convergent technology with the amal-

gamation of current tools for improving efficiency, usefulness, and wide acceptance by enhancing the compatibility of technologies, integrating various tools, and increasing connectivity, trust, and accessibility. These two approaches are defined as follows:

- Climate technology: refers to those technologies that were developed to reduce GHGs. DTs using AIoT, and machine learning can improve energy saving and monitor thermal energy consumption, waste heat, temperature, humidity, and light levels for maintaining occupant comfort and optimizing building operating costs.

- Resilient technology: refers to those technologies that are developed for the continuity of business during uncertain times or pandemics such as COVID-19. This can be a convergent technology that enables a business to have a minimum level of services and requirements during the unpredicted time. While the communication tools for office work and meetings were widely used during the pandemic, a high level of complexity was involved in continuing the construction operation and performance by using remote control technologies during the pandemic. This is an open question to the current demand of the construction business. However, answering this question needs further investigations examining various technical solutions, assessing the reliability and safety of these solutions, and offering many use cases to practitioners.

- Convergent technology: refers to a novel combination or integration of technologies that have the potential to enhance construction performance and operational tasks. The construction industry operates very similarly to how it did many years ago in most activities, and digital tools are used as single forms, mainly disconnected from other systems. The combination or integration provides an opportunity to develop innovative solutions to address various industry challenges. While previous literature was focused on single technology development, there is a long way to improve the practice of integrations and combinations. Shirowzhan et al. [149] discussed that compatibility and interoperability are key challenges that should be considered for convergent technology development.

Some challenges of convergent technology development and readiness are summarized as follows.

o    Autonomous systems and machine-to-machine (M2M): integrated platforms with data exchangeability have not been fully extended and utilized. This helps construction laborers avoid dust and vibrations;

o    Skilled labor and human–machine interface (HMI): The industry needs skilled laborers to be familiar with machine languages with an understanding of data and robots. HMI will be a concern in terms of efficiency and safety;

o    Web of Things (WoT): while the Internet of Things (IoT) is accepted for use in construction projects, there are interoperability challenges among various IoT platforms and standards. This makes the data exchange challenging, and the project managers may need to use different platforms that decrease efficiency. The concept of WoT suggests connecting construction tools and equipment to the web, and the construction manager can detect efficiency and productivity under an online platform. This offers a high level of connectivity, real-time communication of objects, wireless asset trackers, smart health monitoring systems, and autonomous construction equipment where applicable.

## 9. Conclusions

This paper aimed to examine the current literature on convergent technology with a focus on identifying future directions, and to discuss key themes for future investigations. The six objectives were achieved in different sections of the paper. A systematic review method was conducted to identify relevant papers, evaluate the current literature, and feed the discussion of future construction. The outcome of the search was used to create the database required for the content analysis. A set of six keywords were selected to search the Scopus database, which resulted in identifying 289 scholarly peer-reviewed papers related to FOCIT. Based on the established FOCIT dataset, the bibliography analysis

was conducted, and selected papers were used for content analysis. The content analysis helped in discussing the strategic horizon of digital technologies, visualization techniques, and a technology implementation roadmap. The convergent technology for construction purposes can include 4D printing, cloud-based augmented/virtual/mixed reality, Metaverse, AI-integrated IoT, autonomous digital twin systems, automatic guided vehicles, and telematic devices. A popular example of a general convergent technology is a smartphone combining a few unrelated technologies developed by different industries, such as a camera, telephone, game device, music player, and many other tools. The acceptance rate of a useful convergent technology would be much higher than a single technology that barely addresses one need or task in construction. This paper tends to direct future efforts from single technology applications to multi-integrated systems addressing a set of practitioners' needs.

The future of construction is involved in implementing a set of technologies such as DTs, BIM, AI, BCT, and VR. This refers to 'convergent technology' encouraging scholars and innovators to focus on offering novel combinations of various technologies to address current solutions or enhance processes and methods. For a long time, the literature focused on identifying various applications for VR or AR as a single or core technology. AR and VR applications are still limited to some practices and have not been taken seriously by practitioners. However, it can be used as a core part of a digital twin or metaverse, so this makes mixed-reality technology more useful. Practitioners and smaller firms are not able to adopt a wider range of single technologies and learn how to operate them. However, it is more efficient to use a convergent technology as a novel combination of BIM, AR, and BCT connected to a DT. The concept of convergent technology opens the door for investigation and innovation in the construction field.

The convergent technologies can also address climate and resilient needs. The FOCIT literature shows limited technologies focused on these needs, while the concept of convergent technologies has the potential to address climate and resilient needs in construction. The literature on these needs should be extended further since the industry is committed to global climate agendas, and there is a high demand for resilient technology.

The recent disruption, such as COVID-19, revealed that there is a need to enhance the resilience of the construction industry for the continuation of business during an uncertain time. This increased the need for convergent technologies to enable remote work in the future. Although the pandemic negatively affected the industry, the rate of digital communication technology uptake has increased significantly. This suggests that the industry is in a better position in terms of technology readiness and upskilling practitioners in terms of using digital technologies. The present paper suggested nine themes as focuses of future directions. However, future investigations are required to collect relevant grey literature, including government agendas and industry reports to include the policymakers' opinions. The theoretical implications of this paper are to provide a set of clear directions that can be used for structuring a survey to collect practitioners' opinions about their future needs and expectations that a convergent technology might be able to address. The practical implications of this paper are to give directions to practitioners and construction business managers to develop strategies for embracing new technologies, such as developing required infrastructure and upskilling and training their employees.

Considering "construction technology adoption" concepts can assist in predicting and fostering the uptake of convergent technologies to improve efficiency, increase collaboration, enhance quality, and enhance safety by exploiting the advantages of digital tools and data. However, the technology adoption framework is complex and should be defined based on the context considering technical, individual, social, organizational, and environmental factors. Further, the lack of skilled workers, resistance to change, data security issues, complexity in the application, and interoperability issues stifle the uptake of single technologies and can be addressed by utilizing convergent technologies. Future research studies should empirically and analytically examine real scenarios for convergent



technology adoption in various countries. Case studies are known as valuable documents in terms of lessons learned in specific contexts.

Amidst implications and contributions, this study encourages the combination of Big 11 technologies, known as convergent technologies, to assist construction firms in their contractual commitments.

**Author Contributions:** Conceptualization, all authors; methodology, S.M.E.S. and S.S.; software, J.G.R., A.A.K., K.S., X.S. and J.G.R.; validation, all authors; formal analysis, A.A.K., K.S., X.S. and J.G.R.; investigation, all authors; resources, A.A.K., K.S., J.G.R. and X.S; data curation, A.A.K.; writing—original draft preparation, S.M.E.S., A.A.K., K.S., X.S. and J.G.R.; writing—review and editing, all authors; visu. All authors have read and agreed to the published version of the manuscript.

**Funding:** This research received no external funding.

**Data Availability Statement:** All data used in the study is mentioned in the paper.

**Conflicts of Interest:** The authors declare no conflict of interest.

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
