# Peer review of "BIM and Digital Twin for Developing Convergence Technologies as Future of Digital Construction"

_buildings, doi:10.3390/buildings13020441_

Round 1

Reviewer 1 Report

Comments

In this paper, the authors presented a systematic literature review for BIM and Digital twin related technology convergence considering the circular economy. The paper aims to investigate state of play and presented a roadmap on the subject matter. Specific contributions include identifying gaps and eleven disruptive technologies to provide a framework for policymakers. Overall the paper is very nicely written, the authors have done good work and should be accepted for publication as it is. The reviewer has no specific comments to further enhance the quality of the paper.

Few comments that may be considered if deemed necessary by the authors are as follows:

  1. In the science mapping keyword analysis of retrieved articles, the reviewer is confused about how the clusters are defined in the VOSviewer? For instance, how the authors come up with the title of the clusters and why five clusters were formed?
  2. The title include the word “circular economy” but the reviewer thinks its very minute portion of the paper and the title gives another impression that the entire paper revolves around the circular economy. Hence, the title could be revised.
  3. According to the opinion of the reviewer, the topics like climate technology, resilience technology seems like off topics and looks out of scope of this paper but are considered in the last.

Author Response

Respected Reviewer,

Please find attached a word document for the addressed comments.

Regards,

Manuscript authors

Reviewer 2 Report

Dear Authors, 

Good day to you. The article was an excellent review of in-depth research. The literature review was thorough, the methodology was painstakingly specific, and it incorporated the sufficient number of related articles. The discussion part is also good and readable. 

Best Wishes

Author Response

(The authors gave the same response as above.)

Reviewer 3 Report

Manuscript Number: buildings-2165950 

In this review, the authors have first enumerated many new technologies suspected to enhance future digital constructions, briefly introduced their nature, and finally highlighted the recent advance in the circular economy. Many data science and economic modeling research topics match the journal's scope well. Nonetheless, the quality of this manuscript should be improved, and the following issues should be appropriately addressed before consideration for publication.

1. I advise the authors to give a more comprehensive introduction and discussion focusing on a few representations in more detail rather than just enumerating too many examples without a spotlight on current (if any) applications.
2. There are some concerns about the figures. Figure 8: where was it reprinted from? Provide the data source and reprint permission. Figure 7; the whole life cycle cannot be attributed to the specific construction stage at the horizontal level in the timescale. Figure 6. Do the occurrences correspond to the font size? If so, provide the scale bar of such correspondence. Fig.4. Are only the raw materials representing horizontal integration tights?
3. The conclusion is simple and lacks insightful perspectives in this research field. For example, can the authors give some personal opinions about the advantage and disadvantages of using different types of technologies compared with conventional ones?

4. For the review article to make more impact, the shortcomings/drawbacks should be identified for each tool and included in the review. E.g., are now or will there be any law restrictions concerning general safety and personal life protection in using drones, as shown in fig. 9b?

  

Author Response

(The authors gave the same response as above.)

Round 2

Reviewer 3 Report

The review can be accepted